# Influences of Age, Sex and Smoking Habit on Flavor Recognition in Healthy Population

**DOI:** 10.3390/ijerph17030959

**Published:** 2020-02-04

**Authors:** Immacolata Cristina Nettore, Luigi Maione, Silvio Desiderio, Emma De Nisco, Fabiana Franchini, Giuseppe Palatucci, Paola Ungaro, Elena Cantone, Paolo Emidio Macchia, Annamaria Colao

**Affiliations:** 1Dipartimento di Medicina Clinica e Chirurgia, Università degli Studi di Napoli Federico II Via S. Pansini, 5–80131 Naples, Italy; immacolatacristina.nettore@unina.it (I.C.N.); luigi.maione@unina.it (L.M.); silvio.desiderio@gmail.com (S.D.); emmadenisco@gmail.com (E.D.N.); fa.franchini01@gmail.com (F.F.); peppe088@tiscali.it (G.P.); colao@unina.it (A.C.); 2Istituto per l’Endocrinologia e l’Oncologia Sperimentale “Gaetano Salvatore”, Via S. Pansini, 5–80131 Naples, Italy; pungaro@ieos.cnr.it; 3Dipartimento di Neuroscienze e Scienze Riproduttive ed Odontostomatologiche, Università degli Studi di Napoli Federico II Via S. Pansini, 5–80131 Naples, Italy; elena.cantone@unina.it

**Keywords:** flavor, gustometry, retronasal smell, age, food choice, sex, reference values, cigarette smoking, healthy population

## Abstract

(1) Background: Flavor is one of the main factors influencing food preferences and dietary choices, and a reduction in flavor recognition has been associated with several diseases. A novel quantitative test to assess flavor has been recently developed and validated. The aim of the present work was to define the standard of flavor recognition in the general healthy population. (2) Methods: Three hundred and forty-eight healthy volunteers (18–80 years) performed the flavor test (FT). The test consisted of the oral administration of aqueous aromatic solutions, identifying 21 different compounds. Flavor score (FS) was calculated as the sum of the properly recognized flavors (range 0–21). (3) Results: Normal ranges for FT were produced. Flavor recognition was found to decrease with age. Females obtained slightly higher scores than males, mostly at older ages. Cigarette smoking seemed not to influence flavor recognition. (4) Conclusion: The normal values found for the flavor test in the healthy population will allow its usage as a diagnostic tool in several diseases.

## 1. Introduction

The major determinant of food choice is hunger; however, in the presence of different food options, the choice of what we eat is not determined solely by physiological or nutritional needs. Gender, age, education level, and emotional motivations should also be taken into account [1]. In addition, taste perception and food hedonics are also important determinants in such choices [2,3,4]. The experienced pleasure of eating particularly highly palatable foods (sweet and fatty foods) makes individuals more likely to consume them again [5]. Although the molecular mechanisms of such a rewards process are still unclear, it has been suggested that the release of dopamine into the *nucleus accumbens* is modulated by the consumption of highly palatable foods [6]. 

Flavor is a multisensory perception [7], which can be considered a distinct and functional sensory system, as suggested by behavioral [8] and neuroscientific [9] studies. Flavor is determined by the unified perceptual experience or “Gestalt” of a food that arises from the integration of retronasal olfaction (i.e., signals deriving from the retronasal smell through oral cavity) with several peripherally distinct sensory inputs, including taste, texture, viscosity, temperature, sight, and even the sound of foods or oral nociception (pain) [10,11]. Retronasal olfaction is probably the main determinant of flavor detection and enjoyment [12,13] and thus represents the main contributor in the hedonic response and the “pleasantness” of foods [14,15,16]. Animals have innate sensitivities to accept or reject foods, and acquired feeding responses depend on the orosensory and post-ingestive results of foods. Animals prefer the flavor of foods and fluids because these are associated with positive post-ingestive nutritional results [17].

Recently, a novel quantitative test to assess flavor has been developed and validated in normosmic and anosmic subjects [18]. The test is able to combine information on smell through the retronasal pathway, providing integrated information with the taste through the oral cavity. The test has been used to investigate the flavor recognition capacity in patients with endocrine [18] and neurological diseases [19]. By means of this test, it has been demonstrated that flavor is severely impaired in patients with Kallmann syndrome and in patients with Parkinson’s disease.

The aim of the present work was to define the standard of flavor recognition ability in a sample of the general healthy population and to correlate the flavor scores with age, sex or smoking habits.

## 2. Materials and Methods

### 2.1. Subjects

The present study was approved by the Local Ethics Committee (Comitato Etico Università Federico II, ref. no. 253/13). The screened population was composed of volunteers who freely agreed to participate to the study. All the participants were adults (≥18 years of age) who gave their written informed consent, in keeping with Italian Bioethics Law and the Declaration of Helsinki. Five hundred and eight volunteers were recruited in the outdoor hospital of the “Campus Salute Onlus” held in several locations from October 2014 through February 2019 (www.campussalute.it). All the included subjects were part of a project to investigate the role of lifestyle in preventing chronic diseases. The study is therefore a part of a large database started in 2010 to investigate the health status of the general population of Campania Region inhabitants. Subjects with current chronic rhinosinusitis (with or without polyps) and chronic rhinitis as well as subjects reporting or diagnosed with a reduced taste or smell perception were not included in the study. Similarly, subjects reporting chronic diseases (type 1 or type 2 diabetes, cardiological diseases, stroke or other neurological diseases, gastrointestinal problems), and underweight (BMI < 18) or obese (BMI > 29.9) individuals were not included in the study. Subjects receiving drugs known to interfere with taste or smell abilities (antithyroid drugs, antibiotics, griseofulvin, lithium, penicillamine, procarbazine, rifampin, anxiolytics, antipsychotics, antiepileptic drugs, antidepressants, amiodarone, digoxin, statins, and chemotherapeutic drugs) or subjects which reported the abuse of inhalant drugs or substances (i.e., cocaine) were also excluded from the study. The study group ultimately included 348 subjects (F = 241, M = 107), with a mean age of 42.41 ± 15.63 years (median = 41.79 years). All the subjects performed the flavor test, and 136 of them also performed the gustometry assessment. 

### 2.2. Self-Assessment Questionnaires

Prior to the evaluation of chemosensory functions, all the participating subjects were invited to rate subjective chemosensory function, namely flavor (“How would you rate your fine taste, e.g., during eating and drinking?”) and taste perception (“How would you rate your basic taste: sweet, sour, salty, bitter?”) by using a visual analogic scale ranging from 0 (“no taste/flavor perception”) to 10 (“excellent flavor/taste perception”).

### 2.3. Gustometry

The basic gustatory test has been previously described [20]. Umami taste was not included in the study as, in the Italian population, it is often under-recognized or described as a different taste quality [21]. Four liquid taste solutions were used: 1) 10 g/L (29 mM) sucrose for “sweet”; 2) 324 mg/L (1 mM) quinine hydrochloride for “bitter”; 3) 1.0 g/L (0.017 M) sodium chloride for “salty”; and 4) 0.2 g/L (1 mM) citric acid for “sour”. These substances were dissolved in distilled water, and a drop of approximately 20 µL of each solution was applied on the upper surface of the tongue. Before the application of each taste solution, the mouth was rinsed twice with distilled water. After the presentation of the stimulus, each subject was invited to choose one of the four descriptors (“sweet”, “sour”, “salty”, “bitter”). Each solution was applied four times in a pseudo-randomized order. The gustometry score (GS) was calculated as the sum of correctly identified tastes and ranged from 0 to 16.

### 2.4. Flavor Test

The flavor test was developed and patented (patent no. 0001426253, category A61B500 of the Italian Ministry for the Economic Development) and applied in a group of 70 normosmic and 54 anosmic subjects [18]. Its performances compared to the standardized threshold discrimination identification (TDI) score obtained by Sniffin’ Sticks [22]. It consists of a series of 20 aromatic extracts, corresponding to routine dietary use flavors for Italian people: almond, banana, cheese, chocolate, coffee, fish, garlic, green mint, hazelnut, honey, lemon, licorice, mushroom, mustard, onion, peach, roasted beef, smoked, tea and vanilla. The aromas were kindly provided by the manufacturer GIOTTI (Enrico Giotti spa, Scandicci, Firenze, Italy). Each flavor was diluted as previously described [18], according to the manufacturer’s instruction. An aliquot of 0.5 mL of each flavor was administered in the oral cavity and left for approximately 5 seconds. The mouth was rinsed twice with distilled water before the administration of the following flavor. At each administration, participants were invited to identify the aromatic elements by making a choice out of five proposed items. A total of 21 aromatics (including one blank: water) were administered. The flavor score (FS) was calculated as the sum of correctly identified aromatics and ranged from 0 to 21 [18].

### 2.5. Statistical Analysis

Statistical analyses were performed using MedCalc 14.0 (MedCalc, Ostend, Belgium) software and the Statistical Package for the Social Sciences Software Version 25 (SPSS v.25, IBM, Armonk, NY, USA). Results were expressed as means ± standard deviation (SD) for continuous variables and as frequencies for categorical variables. The Kolmogorov–Smirnov goodness of fit test was used to assess the hypothesis of a normal distribution of data. Most variables did not have a Gaussian distribution; thus, non-parametric tests were used for group comparisons (Kruskal–Wallis test). Correlations were calculated with the Spearman’s correlation coefficient. Fisher’s exact test was used to compare categorical data. The level of significance was set at α = 0.05, with a two-sided level.

## 3. Results

### 3.1. Population Characteristics

The 348 included subjects had a mean age of 42.41 ± 15.63 years (median = 41.79 years). Females (n = 241) represented 69.3% and males (n = 107) were 30.7% of the studied population. The median and distribution of ages were not different between males and females. The BMI, in the studied population, ranged from 18.0 to 29.8 with a mean of 23.85 ± 3.00 kg/m2. Non-smokers were considered to be subjects that never smoked or that had stopped smoking at least ten years prior to the test. The number of smokers in the sample amounted to 83 (23.9% of the total), with 55 females (66.3%) and 28 males (33.7%). No current or previous smokers of tobacco or flavored-electronic cigarettes were found in our study. Smokers smoked an average of 10.52 ± 8.20 cigarettes/day, and the smoking habit lasted on average 15.15 ± 12.77 years. BMI distribution was the same between smokers and non-smokers. Sex and smoking habit were not different between the enrolled subjects, when population was divided into two groups according to the median age. The characteristics of the studied population are shown in Table 1. Since only a subgroup of subjects performed both the flavor test and gustometry test, the data in the table are shown as separate columns.

### 3.2. Self-Assessment Questionnaires, Flavor Score and Gustometry

Subjective chemosensory function was investigated for the whole population. Self-evaluated flavor ranged from 3 to 10 with a mean of 7.53 ± 1.51, while self-evaluated taste ranged from 3 to 10 with a mean of 7.98 ± 1.56. 

In the overall population, the flavor score (FS) ranged from 5 to 21 with a mean of 14.47 ± 3.06, and the mean gustometry score (GS) of the 136 subjects performing gustometry was 14.14 ± 2.00. No significant correlation was determined between FS and GS.

### 3.3. Age Effect

Age was inversely correlated with both flavor (*ρ* = −0.308, *p* < 0.001; Figure 1a) and gustometry (*ρ* = −0.208; *p* = 0.015; Figure 1b). 

### 3.4. Sex Effect

The flavor score was significantly lower in men (13.98 ± 2.92) than in women (14.69 ± 3.10) (*p* = 0.021) (Figure 2a). FS was inversely correlated with age, which remained true when the population was divided according to gender (F: (*ρ* = −0.256, *p* < 0.001; M: *ρ* = −0.404, *p* < 0.001). When the population was split into two groups according to the median age, differences between males and females were significant (*p* = 0.018) only in the group of older subjects (>41.79 years) (Figure 2b). No significant differences were determined in terms of the GS between men and women.

### 3.5. Smoke Habit Effect

No differences were observed in FS or GS between smokers and non-smokers when evaluating either the whole population (Figure 3) or the population split into two groups based on the median age (Appendix A) or sex (Appendix A). Similarly, the number of smoked cigarettes and the accumulated smoked years (corrected for age) did not influence FS or GS (Appendix A).

### 3.6. Flavor Score Reference Values

In Figure 4a–b, distributions, medians and fifth and 95th percentiles of FS split based on sex and age are shown. Accordingly, normal ranges of FS were calculated as a reference for future tests (Figure 4c–d).

### 3.7. Comparison between Flavor Score and Gustometry with Self-Assessment Questionnaires

No correlation was found between the self-assessed degree of flavor recognition and FS (*ρ* = 0.155; *p* = 0.076). A weak but significant positive correlation was found between measured FS and self-evaluated taste recognition scale (*ρ* = 0.150; *p* = 0.005). GS did not correlate with either flavor or taste self-assessed evaluations (*ρ* = −0.95; *p* = 0.275; *ρ* = 0.107; *p* = 0.217, respectively). This could be the consequence of the reduced number of subjects performing GS. Sex did not influence the self-assessed scores for basic taste perceptions. Female had a slight but significantly higher (F = 7.63 ± 1.54; M = 7.29 ± 1.4; *p* = 0.021) self-estimated flavor perception than men (Appendix A). No gender differences were detected in self-estimated basic taste perception (Appendix A). Self-assessed perception of flavor and basic taste was not influenced by age (Appendix A). 

### 3.8. Difference in Flavor Recognition: Individual Tastants

No significant differences in specific flavor recognition were detected when comparisons were made according to gender or smoking habit (after Bonferroni’s correction for multiple tests) (Appendix A). 

## 4. Discussion

In this work, flavor recognition has been investigated using the flavor test—a specific test to assess the retronasal olfactory function—in 348 healthy volunteers. Part of this population was also tested for their taste ability.

A decline in smell identification and in gustatory function is common in the elderly [23,24,25,26]. It has been hypothesized that this might be determined by the combination of several factors, including a reduction in chemosensory perception, poor oral health and changes in olfactory function [25]. Previously published studies investigated the effect of age on basic taste. Most studies demonstrated that aging may produce a decline in gustatory function [25]. On the other hand, Croy et al. found no association between retronasal olfaction and age [27], although this work suffered from at least two limitations, namely the limited number of subjects and the different ethnic and cultural context of the examined population. The results of the present work indicate a negative correlation between age, flavor recognition and taste functions in a large number of healthy subjects belonging to the same genetic and cultural background. The nutritional habits of the studied population were also homogeneous (see Appendix A). The correlation between gustometry and flavor scores seems to suggest that chemosensory sensitivity could be a critical issue. 

Several lines of evidence indicate a gender-related ability in taste function in humans [28,29] as well as in rodents [30,31]. Sex hormones can alter taste processing as well as taste-related behaviors [32]. In general, women seem more able to detect basic taste stimuli when compared to men [33]. Anatomical data also support the sex difference in taste recognition, with women having more fungiform papillae and more taste buds than men [28]. Many studies have supported superior female performance also in odor identification [34]. Reports specifically investigating the retronasal olfactory function indicated sex-related differences, with women having higher scores than men [27,35]. The results of the present study are in the same direction, suggesting that females attain slight but significantly higher FSs than males. Interestingly, these differences are particularly clear in older subjects (>41, 79 years). Therefore, the improved ability in women should not be merely attributed to estrogens, since it does not fade with advancing age. Women have a greater amount of taste receptors on their tongue, and the reason for this seems to be closely linked to evolution. They would therefore be evolutionarily better equipped to distinguish dangerous, spoiled or poisonous foods. Hence, they decide whether the food they consume is healthy for themselves and for their children. For the same reason, they also have a better sense of smell [28]. Moreover, we could speculate that gender-related flavor and taste recognition differences may not only have biological bases but may also be related to social components. Indeed, in the examined population, females were most likely to be involved in alimentary choices and food preparation, and therefore they probably had more experience and training with odors and flavors than males [36]. 

The effects of smoking habit on taste or flavor are still not univocal [37,38,39,40]. A recent systematic review pointed out the level of controversy of the existing studies on this topic [41]. The results of the present study suggested no differences between smokers and non-smokers in flavor and taste recognition, as well as in specific taste identification. These data should however be carefully considered, since the smoker population was very limited and restrained to 83 subjects (24% of the total). Nevertheless, our observations are in line with a recent report indicating that the number and the morphology of fungiform and filiform papillae showed no significant differences between smokers and non-smokers and that taste function is similar between males and females [42].

Finally, it is interesting to note that, in contrast to reports in previous studies [43], we found stronger discrepancies between self-assessed and measured scores for flavor and taste. Although self-reported scores to evaluate smell capacity have been used, and the reported data suggested the good sensitivity of these techniques [43,44], our results indicate that subjects with reduced flavor or gustometry scores are not aware of their own disabilities. It is widely accepted that age determines a decline in flavor perception and pleasantness [45]; however, this is normally compensated through the use of improved visual cues, easily recognizable foods, and/or identity labels [46]. Our test prevented the use of any additional stimulus that could have helped the flavor recognition. In addition, we cannot exclude that part of the discrepancy is due to the lack of a comparison target, which has been shown to increase the self-reported olfactory function accuracy in other studies [47]. Since the main aim of this work was not the comparison of the self-reported capacities with flavor score and gustometry, we used a simple analogical 0–10 scale. We failed to show any correlation between GS and either flavor or taste self-assessed evaluations. It cannot be excluded that the smaller number of subjects performing gustometry could have impacted these results.

Some limitations of this study should be highlighted. First, the recruitment was performed at several outdoor hospitals within a project to investigate the role of lifestyle in preventing chronic diseases. This probably determined an unbalance in the selection of the studied population in terms of age, sex and smoking habit. Moreover, most of the information on patients’ health characteristics was obtained from a simple interview, and the collection lacked any medical visits or biological samples. A larger intervention study is necessary to optimize the selection of the studied subjects to confirm the results reported here. In fact, since the test has been used for the first time in a normal population and no previous reference studies are available, it was not possible to calculate the power of the study and the adequacy of the sample. In addition, gustometry was not explored in all the subjects, and no thresholds have been calculated for each of the basic tastes. This information would have been useful to increase the clarity of the results. Moreover, the gustometry investigation did not include the umami taste. Although umami is now considered one of the five basic tastes, in the Italian population, this taste is often under-recognized or described as a different taste quality [21]. Finally, we are aware that the data presented here are based on a single population, and that conclusions may be different with a larger or different human population.

## 5. Conclusions

Our study evaluates flavor recognition abilities in a large healthy population using a recently developed sensorineural test. The results obtained allowed us to calculate FS normative values according to age and sex. These values could be potentially used as reference values to evaluate flavor recognition alterations in various pathologic conditions. This study confirms that age is an important factor influencing flavor recognition as well as that females are more able than males in flavor recognition, especially at older ages. Finally, cigarette smoking did not seem to influence flavor recognition.

The test used to evaluate flavor recognition is simple, quick and allows us to screen a large number of individuals at a time. We propose the application of the flavor test as a potential screening test in several clinical situations that may be associated with a decrease in odor or taste perception. Moreover, the test can potentially be used in healthy population studies to assess how the retronasal olfactory sense influences food choices. 

## Figures and Tables

**Figure 1 ijerph-17-00959-f001:**
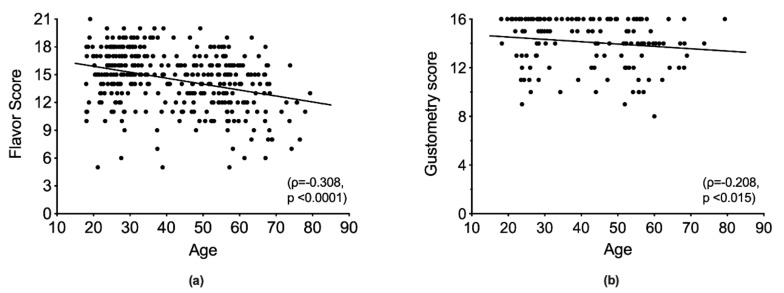
The effect of age on flavor score (**a**) and gustometry (**b**). Age was inversely correlated with both the flavor score (*p* < 0.0001) and gustometry score (*p* = 0.015).

**Figure 2 ijerph-17-00959-f002:**
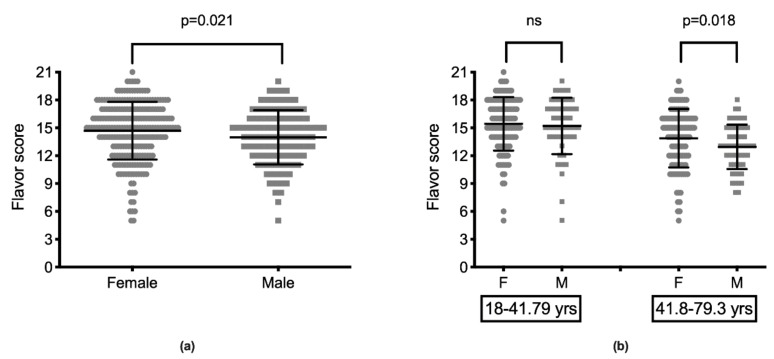
Sex differences in flavor recognition. Mean + SD of flavor score (FS) (**a**). Females (dots, n = 241) had significantly higher FSs than males (squares, n = 107). When the studied population was split into two groups according to the median age (41.79 years) (**b**), no differences between females (n = 127) and males (n = 49) were present in younger subjects, while FSs were significantly higher in females (n = 114) than in males (n = 58) in older subjects.

**Figure 3 ijerph-17-00959-f003:**
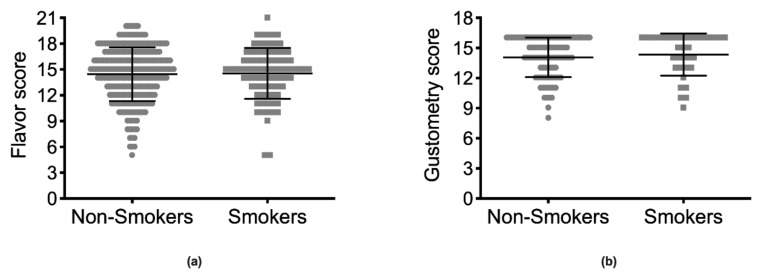
Flavor and gustometry scores according to smoking habit. Mean ± SD of the flavor score (**a**) and gustometry score (**b**) for non-smokers and smokers. No significant differences can be determined between the two groups for either flavor or gustometry scores.

**Figure 4 ijerph-17-00959-f004:**
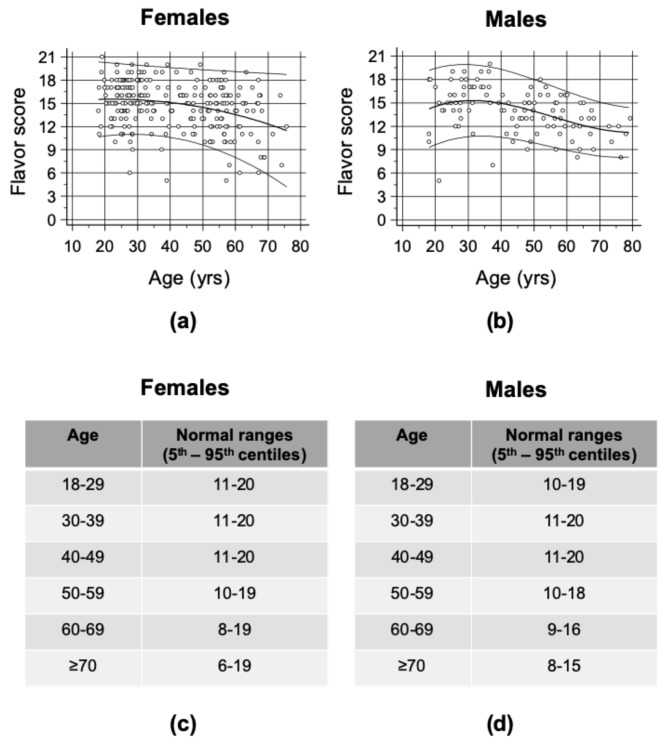
Age-related reference interval for flavor score. In the figure, the flavor scores of all the examined females (**a**) and males (**b**) and the age-related reference intervals (**c**,**d**), considering normal values of FS between the fifth and the 95th percentiles, are shown. Lines correspond to the mean (bold line) and to the fifth and the 95th percentiles (thin lines).

**Table 1 ijerph-17-00959-t001:** Characteristics of the studied population.

	Flavor	Flavor + Gustometry	*p*-value
No. of subjects included	348	136	
Age	42.41 ± 15.63	41,18 ± 15,70	Not significant
Sex F/M (%)	241/107 (66.3/33.7)	96/40 (70.6/27.4)	Not significant
BMI	23.85 ± 3.00	23,89 ± 3,01	Not significant
Smoking habit (No/Yes, %)	75.6/24.4	74.4/25.6	Not significant
No. of smoked cigarettes	10.52 ± 8.20	9.13 ± 7.06	Not significant
Years of smoking	15.15 ± 12.77	15.94 + 14.85	Not significant

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
