# Peer review of "Influences of Age, Sex and Smoking Habit on Flavor Recognition in Healthy Population"

_ijerph, 2020, doi:10.3390/ijerph17030959_

Round 1

Reviewer 1 Report

This is an interested study on flavor recognition, based on gustometry and flavor test scores in Italian population. This study is of interest, clearly organized and well described, with novelty aspects  and it needs minor correction before accepted for publication. Self-estimated flavor perception scores might be presented in 3.2 section.

Reviewer 2 Report

This paper described the application of a flavour recognition test and its validation on a generally healthy population. Some points need clarification.

Introduction

In general, more information regarding the intrinsic relationships between flavor and hedonic acceptability of food products should be included. This is important since one of the justifications in this study is to apply a flavour recognition test to the general population as a link to understand overall dietary choices and food acceptability. Moreover, there should be more information regarding the physiological differences in humans as a variable factor in the recognition of different flavours and aromas.

Line 38: Is there any reference that can support this statement?

Discussion

There should be information regarding the persons that are considered “super-tasters” or that they have an increased ability to perceive specific tastes and flavour compared to the general population. There are several works in the sensory analysis area regarding the differences between “super-tasters” and the regular population that can be stated in this section.

By clustering population in their tasting abilities (including flavour), more meaningful analysis can be done to differentiate food choices and behaviours.

Reviewer 3 Report

The authors present data on the relationship between gustatory and flavour scores with potential modifying factors and claim to present reference/normative values for a healthy population. I do not think the research itself is flawed, but there are many flaws in the way the data and background are presented and interpreted that need to be rectified before this manuscript could be considered for publication.  

Title - Flavor recognition in healthy population – is very vague and does not reflect the body of work presented. This should be revised.

Introduction

Many references presented in the introduction are not reflective of the statements they are linked to. Many of these are catalogued below and I encourage authors to check the suitability of these and all their references.

In developed societies where food is readily available, food preferences are the first factor  influencing food choices [1].” – authors should clarify what they mean by “first” factor, is it the first consideration or the primary consideration? They should also clarify how they are defining preference. Furthermore, the reference given does not appear to support the statement – the results from the survey in the weblink do not show this they show that Taste is the most important, which is not the same as preference, followed price, healthfulness and convenience and sustainability. There are many proper academic references that could be used to support such an argument that are more suitable than a self-published survey. The reference also does not support using the phrase “developed societies” as it was a survey of AMERICANS only.

Dietary preferences are determined by physiological, social, psychological [2] and genetic factors [3]  – reference 2 is not appropriate to this statement as a distinction needs to be made between food choices and preferences and this study was using in depth qualitative interview to determine how people reconciled different values when making food choices, it was not measuring factors in determining preference – this reference appears to have been chosen based on information provided in the introduction of this study, not the outcomes of the study itself. Reference 3 is also not wholly appropriate as it is on taste genetics only, and the authors, many other genetic factors are involved in food choice, preferences and intake.

The statement “however flavor rather than taste should be considered the main factor modulating food preferences” requires a reference or needs to be reframed as an hypothesis if this is not yet shown.

It would be preferable if the references to the books in references 8 & 9 were replaced with relevant peer reviewed publications

The definition of retro-nasal olfaction provided “(i.e. signals deriving from the retronasal smell through oral cavity)” should be provided at the first use of the term not the second.

“Retronasal olfaction (i.e. signals deriving from the retronasal smell through oral cavity) is nonetheless the main determinant of flavor detection and enjoyment” it is unclear how reference 14 supports this statement as retronasal olfaction and taste were assessed in this reference, not texture, viscosity, temperature, sight, sound or nocioception, which are all listed in the previous sentence to which the “nonetheless” refers. Reference 15 supports retronasal olfaction as being important but only assess it relative to self-reported taste and flavour perception, which in no way demonstrates retronasal olfaction being the main determinant. I’m not disagreeing that retronasal olfaction is the most important determinant, but authors should provide appropriate references to back up that statement.

Regarding reference 19 (also used in the methods), authors should explain clearly what they mean when they say the flavour test has been “validated”. Especially as they state in the results that “No correlation was found between the self-assessed degree of flavor recognition and FS (r = 173 0.155; p = 0.076).”

Methods

More details and/or a reference should be provided on the self-assessment questionnaire, how was the scale presented, how were the questions were phrased? What foods listed to self assess flavour or was the question just on foods in general? Importantly the self assessment questionnaire and the gustatory tests are listed as including the basic tastes sweet, bitter, salty and acid… are the authors denying the definition of umami as a basic taste? Most in the taste field would disagree with this, umami has be internationally recognized as a basic taste since the 1980s.

A symbol appears to have been missed in writing or formatting, the methods state 20 L of solution was applied to the tongue, which is clearly not correct. I’m assuming 29 microliters?

Results

Population characteristics are describe and comparisons made between males and females and included/excluded participants, and by median age. But only a subset (136 subjects) completed the gustomtry testing. Therefore the population characteristics for those undergoing this testing vs those who did not should also be presented. I suggest presenting the full population characteristics and comparison of relevant sub categories in tabular form.

The rationale of the “slicing and dicing” in the presentation of the sex effect are unclear and authors should provide a rationale for this approach in the methods, why not adjust/correct for age in the model instead of splitting my median and then splitting by age.

The smoking anaylsis is presented as the whole cohort and split by median age, but the sex effect reported in the previous section is not consided here. This should be rectified.

Similarly, the number of smoked cigarettes and the cumulated smoked years 155 (corrected for age) did not influence FS and GS (data not shown).”  Data not shown is unacceptable if authors want to report the finding, they should present the data, even if it is a supplementary materials section. Either show the data or remove the statement.

“Considering age and sex effects, we were able to obtain the reference values for the FS.” A clear explanation as to what authors mean by reference values is required, do they just mean the median and ranges?

“A significant correlation was found between measured FS and self-evaluated taste recognition scale (r = 0.150; p = 0.005). GS did not correlate with either flavor or taste self-assessed evaluations (r = -0.95; p = 0.275; r = 0.107; p = 0.217, respectively; data not shown).”  Given only half the subjects did the GS this is not surprising –this needs to be discussed.

In table 1 21 individual tests are performed with 2 significant results. It is highly likely that this is due to multiple testing and the rate of false positives. A correction for multiple testing should be applied.

“No 183 differences in flavor recognition individual tastants was found between smokers and non-smokers (data no shown).” Again – either show the data (use supplementary materials) or remove the statement, peer reviewers and the scientific community at large can’t just be expected to accept a statement without seeing the numbers.

Discussion and Conclusion

Finally, investigation of flavor perception in general population can  provide clues to the alimentary industries in order to develop foods which are more “pleasant” or better recognized from the population, with potential economic and health interests” This conclusion is not supported by the information presented as pleasantness was not assessed and self assessement of flavour did not correlated with the FS in this study.

“The study demonstrates that age is the most relevant factor influencing flavor recognition.” This statement needs to be tempered, you can say it is more important than smoking as smoking was not significant, you can say it is *an* important factor, but you can’t say it’s the *most important” factor as you also showed sex was important and there are many other factors that may be important that have not been tested. Sex and age would need to be modelled together in order to “rank” their importance. It is also already well established that flavour perception declines with age so it is unclear how this finding is novel (eg. Stevens et al, Appetite Volume 2, Issue 2, June 1981, Pages 127-136, Kremer et al Chemical Senses Vol 32 (6) Jul 2007 pp 591-602) and many others). Authors themselves report “A decline in smell identification and in gustatory function is common in the elderly [22-24].” Is this suggesting that the authors believe that previous studs did not actually assess flavour perception?

A discussion of the power of the study (with particular reference to the sub groups and FS vs GS powers) is required.

Throughout the manuscript the authors focus on the “significance” of the correlations or differences with no reference to the STRENGTH or MAGNITUDE of the associations or difference. This needs to be rectified.  

The conclusion in the abstract is “Reference values for the flavor test in healthy population will allow its usage as diagnostic tool in several diseases.” However ,reference values are not mentioned in the conclusion in the body of the text.

Furthermore, several limitations in the recruitment process are listed – can the claim of producing normative values truly be made in light of these?

Round 2

Reviewer 2 Report

The authors have correctly addressed the comments made by the reviewers.

Author Response

Thank you. A spelling check has been performed

Reviewer 3 Report

From response: “Point 7: “Importantly the self assessment questionnaire and the gustatory tests are listed as including the basic tastes sweet, bitter, salty and acid… are the authors denying the definition of umami as a basic taste? Most in the taste field would disagree with this, umami has be internationally recognized as a basic taste since the 1980s.”

 Response 7: -  We agree with the referee on this aspect, umami is internationally recognized as a basic tastant. However, as we reported in the discussion (lines 281-283) and in the reference 48, this taste is often under or misrecognized in the Italian population.”

If this is the case this should be explained AND referenced in the methods section, not just mentioned in the discussion.

Author Response

Point 1: From response: “Point 7: “Importantly the self assessment questionnaire and the gustatory tests are listed as including the basic tastes sweet, bitter, salty and acid… are the authors denying the definition of umami as a basic taste? Most in the taste field would disagree with this, umami has be internationally recognized as a basic taste since the 1980s.” Response 7: -  We agree with the referee on this aspect, umami is internationally recognized as a basic tastant. However, as we reported in the discussion (lines 281-283) and in the reference 48, this taste is often under or misrecognized in the Italian population.”

Response 1: As suggested by the referee, the sentence has now been inserted in the method section (lines 95-97)